MCH and apomorphine in combination enhance action potential firing of nucleus accumbens shell neurons in vitro

Hopf F Woodward 1 woody@gallo.ucsf.edu
Seif Taban 1
Chung Shinjae 2
Civelli Olivier 3
1 Ernest Gallo Clinic and Research Center, Department of Neurology, University of California , San Francisco, Emeryville, CA , USA
2 Department of Molecular and Cell Biology, Helen Wills Neuroscience Institute, University of California , Berkeley , USA
3 Departments of Pharmacology and Developmental and Cell Biology, University of California , Irvine, CA , USA
Hökfelt Tomas
Electronic publication date: 2013 Apr 9
Publication date: 2013
Volume: 1
Electronic Location ID: e61
Received 2012 Dec 7; Accepted 2013 Mar 12
Copyright: © 2013 Hopf et al.
Copyright year: 2013
Copyright holder: Hopf et al.
License: This is an open access article distributed under the terms of the Creative Commons Attribution License, which permits unrestricted use, distribution, and reproduction in any medium, provided the original author and source are credited.
License URL: https://creativecommons.org/licenses/by/3.0/

Keywords: Schizophrenia, Nucleus accumbens shell, MCH, Dopamine receptor, Apomorphine

Funding: NIDA/NIH DA024746 (OC, FWH) DA028065 (TS) This work was supported by the NIDA/NIH DA024746 (OC, FWH) and NIDA/NIH DA028065 (TS). The funders had no role in study design, data collection and analysis, decision to publish, or preparation of the manuscript.

==============================
The MCH and dopamine receptor systems have been shown to modulate a number of behaviors related to reward processing, addiction, and neuropsychiatric conditions such as schizophrenia and depression. In addition, MCH and dopamine receptors can interact in a positive manner, for example in the expression of cocaine self-administration. A recent report (Chung et al., 2011a) showed that the DA1/DA2 dopamine receptor activator apomorphine suppresses pre-pulse inhibition, a preclinical model for some aspects of schizophrenia. Importantly, MCH can enhance the effects of lower doses of apomorphine, suggesting that co-modulation of dopamine and MCH receptors might alleviate some symptoms of schizophrenia with a lower dose of dopamine receptor modulator and thus fewer potential side effects. Here, we investigated whether MCH and apomorphine could enhance action potential firing in vitro in the nucleus accumbens shell (NAshell), a region which has previously been shown to mediate some behavioral effects of MCH. Using whole-cell patch-clamp electrophysiology, we found that MCH, which has no effect on firing on its own, was able to increase NAshell firing when combined with a subthreshold dose of apomorphine. Further, this MCH/apomorphine increase in firing was prevented by an antagonist of either a DA1 or a DA2 receptor, suggesting that apomorphine acts through both receptor types to enhance NAshell firing. The MCH/apomorphine-mediated firing increase was also prevented by an MCH receptor antagonist or a PKA inhibitor. Taken together, our results suggest that MCH can interact with lower doses of apomorphine to enhance NAshell firing, and thus that MCH and apomorphine might interact in vivo within the NAshell to suppress pre-pulse inhibition.

Introduction

In addition to regulation of feeding, the melanin-concentrating hormone (MCH) system is an important regulator of a number of behaviors related to reward processing, addiction and other neuropsychiatric conditions such as schizophrenia and depression (Georgescu et al., 2005; Nestler & Carlezon, 2006; Shirayama & Chaki, 2006; Chung et al., 2009; Morganstern et al., 2010; Sears et al., 2010; Mul et al., 2011; Chung et al., 2011a; Chung et al., 2011b). The dopamine system also plays a key role in many addiction- and motivation-related behaviors and neuropsychiatric conditions (Di Ciano et al., 2001; Di Chiara, 2002; Nestler & Carlezon, 2006; Shirayama & Chaki, 2006; Di Ciano, 2008; Wise, 2008). Interestingly, dopamine receptors and MCH can interact positively in the regulation of some behaviors including in animal models of cocaine addiction and schizophrenia (Chung et al., 2009; Chung et al., 2011a), while antagonistic interactions are also observed (Georgescu et al., 2005, see also Tyhon et al., 2008).

A recent study (Chung et al., 2011a) examined pre-pulse inhibition (PPI), where a startle response to an intense stimulus can be suppressed when a weak stimulus (the pre-pulse) immediately precedes the intense stimulus. The PPI is used as a measure of sensorimotor gating, and significant deficits in PPI are observed in patients with schizophrenia and some other neurological disorders, which may contribute to sensory overload and related symptoms (Braff, Geyer & Swerdlow, 2001). Chung et al. (2011a) found, in mice, that MCH had no effect alone on PPI but that MCH did enhance the ability of lower doses of the DA1/DA2 dopamine receptor agonist apomorphine to suppress PPI. One implication of this finding is that simultaneous modulation of MCH and dopamine receptors might represent a novel therapy for treatment of schizophrenia, which would allow use of lower doses of dopamine modulators and thus reduce unwanted side effects. However, to fully understand the mechanism of such an interaction, it would be useful to know which brain region these dopamine and MCH agents might interact within, which could suggest additional methods through which neurons targeted by dopamine and MCH could be modulated. Dopamine and MCH receptors are present in many brain regions (Missale et al., 1998; Saito et al., 2001), and thus the region within which apomorphine and MCH might interact to suppress PPI remains unclear.

We previously used in vitro electrophysiology and biochemistry to show that MCH can interact cooperatively with dopamine receptor agonists to increase activity of nucleus accumbens (NAcb) shell (NAshell) medium spiny neurons (Chung et al., 2009). The NAshell is an important regulator of feeding, response to novelty, and some addiction- and reward-related behaviors including suppression of several forms of reward-related responding (Di Chiara, 2002; Georgescu et al., 2005; Chung et al., 2009; Besson et al., 2010; Morganstern et al., 2010; Ghazizadeh et al., 2012; LaLumiere, Smith & Kalivas, 2012). In addition, the NAshell has been noted as a brain region which regulates PPI (Kodsi & Swerdlow, 1997; Alsene, Fallace & Bakshi, 2010), including where dopaminergic drugs can disrupt PPI (Devoto et al., 2012; Meyer & Louilot, 2011; but see Swerdlow et al., 2007), as well as other behavioral models considered to reflect symptoms of schizophrenia such as behavioral inflexibility (Floresco, Zhang & Enomoto, 2009) and disturbances of latent inhibition (Murphy et al., 2000; Weiner, 2003).

Here, we have investigated whether MCH interacts with apomorphine to increase in action potential (AP) firing in the NAshell in vitro, and examined which dopamine receptors might be required for this increase in activity. A cooperative interaction between MCH and apomorphine in vitro could suggest that the NAshell is a brain region where MCH and apomorphine interact to suppress PPI.

Materials and Methods

Brain slice preparation and electrophysiology methods

Male Sprague-Dawley rats (p21-35) were deeply anesthetized with pentobarbital (100 mg/kg, i.p.) in accordance with Institutional Animal Care and Use and National Institutes of Health Guidelines and under the guidance and approval of the IACUC Committee of the Ernest Gallo Clinic and Research Center. Transcardial perfusion was performed with ∼ 15 ml of ice-cold sucrose-aCSF (in mM: 75 sucrose, 87 NaCl, 2.5 KCl, 1.25 NaH2PO4, 7 MgCl2, 0.5 CaCl2, 25 NaHCO3, 1 ascorbic acid) saturated with 95% O2 and 5% CO2. The brain was rapidly removed and coronal slices (300 µm) were cut in this same sucrose-aCSF. Slices were transferred to the recovery chamber containing aCSF (in mM: 126 NaCl, 2.5 KCl, 1.2 NaH2PO4, 1.2 MgCl2, 2.4 CaCl2, 18 NaHCO3, 11 glucose, with pH 7.2–7.4 and mOsm 304-306) at 31 °C and saturated with 95% O2 and 5% CO2. 1 mM ascorbic acid was added to the aCSF in the recovery chamber just before adding the slices. Slices were incubated for 30 min-5 h prior to recordings.

Whole-cell current-clamp recordings were made from NAshell medium spiny neurons, identified as previously described (Hopf et al., 2003). Patch-clamping was performed as described (Hopf et al., 2010a; Hopf et al., 2010b) using 3–5 MΩ glass electrodes filled with a potassium methanesulfonate-based internal solution (130 mM KOH, 105 mM methanesulfonic acid, 17 mM hydrochloric acid, 20 mM HEPES, 0.2 mM EGTA, 2.8 mM NaCl, 2.5 mg/ml MgATP, 0.25 mg/ml NaGTP, pH 7.2–7.4, 275–285 mOsm). Patch-clamp data were collected at 15 kHz and filtered at 2 kHz using Clampex 9.2 or 10 and a Multiclamp 700A patch amplifier (Axon Instruments, Union City, CA). During electrophysiology experiments, slices were submerged and continuously perfused (using a peristaltic pump, ∼2.5 ml/min) with carbogen-bubbled aCSF warmed to 31–32 °C, and supplemented with DNQX (10 µM, to block AMPA receptors) and picrotoxin (50 µM, to block GABA-A receptors). Here, we decided to use a concentration of 2 µM α-flupenthixol, a D1/D2 receptor antagonist. Previous reports have reported divergent concentrations of α-flupenthixol in slice experiments (10 µM in Cameron & Williams, 1993, 0.1 µM in Benardo & Prince, 1982). Also, behavioral experiments injecting α-flupenthixol into the NAcb tend to use higher concentrations (≥5 µg, Di Ciano et al., 2001; Saunders & Robinson, 2012) relative to that used for SCH23390 or raclopride (1–2 µg, see Yun et al., 2004). Thus, we chose 2 µM, which is somewhat higher than the 1 µM we use for SCH23390 and raclopride in vitro. In addition, α-flupenthixol had no effect of firing on its own (Fig. 2C, −0.3 ± 2.3% change in firing versus basal, P = 0.116 paired t-test), in agreement with our previous studies that blockers more selective for D1 or D2 receptors have no effect on firing on their own in the NAshell (Hopf et al., 2003).

Figure 1 Increased NAshell firing with apomorphine and MCH.

(A, B) (A) Example traces and (B) grouped data across time showing that 1 µM MCH (MCH) and 3 µM apomorphine (Apo3) interact to enhance firing in NAshell neurons in vitro, while 3 µM apomorphine had no effect alone. (C) 10 µM apomorphine (Apo10) was sufficient to enhance NAshell firing. (D) 1 µM MCH in combination with 1 µM apomorphine did not increase firing. (E) 1 µM MCH in combination with 10 µM apomorphine significantly increased firing. (F) 3 µM apomorphine in combination with 1 µM MCH significantly enhanced firing under conditions where AMPA and GABAA receptors were not blocked. (G) 3 µM apomorphine + 1 µM MCH did not increase firing when the PKA inhibitor peptide PKI (20 µM) was included in the intracellular pipette.

Figure 2 Dopamine receptor inhibitors prevent the apomorphine/MCH enhancement of firing.

Enhancement of NAshell firing by MCH (1 µM) + apomorphine (3 µM) was prevented by pre-exposure to (A) the DA1 receptor blocker SCH23390 (SCH, 1 µM), (B) the DA2 receptor blocker raclopride (Racl, 1 µM), (C) the D1/D2 blocker α-flupenthixol (flupen, 2 µM), or (D) the MCH receptor blocker TPI (2 µM).

AP generation, analyses and statistics

The resting membrane potential was determined just after breaking into a neuron, and each neuron was then brought to a resting potential of ∼−85 mV by passage of DC current via the patch amplifier before further firing experiments. To generate APs, neurons in current clamp were depolarized with a series of 7 or 8 500-ms current pulses, with 20 pA between each current pulse, and where the initial current pulse for each neuron was just sub-threshold for firing. This series of current pulses was applied every 30 s throughout the experiment, and was alternated with hyperpolarizing steps (−50 and −100 pA) to examine input resistance.

To analyze changes in firing for each cell, a depolarizing step was chosen that gave ∼6–7 APs in the basal condition. We then determined the number of APs at this current step at all time points during the experiment for that cell. The baseline firing rate was determined from the average of firing in the 4 min before addition of apomorphine ± MCH, and was normalized to 100% for each cell. The percent change in firing versus baseline was then determined by taking the average of firing during the 6th–8th min of apomorphine ± MCH exposure. Statistics across groups were determined using these averaged values for the peak change in firing. Thus, a one-way ANOVA was calculated comparing 3 µM apomorphine + MCH with and without dopamine or MCH receptor blockers, and also comparing with 3 µM apomorphine alone; Tukey was used as a post-hoc. P < 0.05 was taken to indicate significant differences. Data are shown as mean + /− SEM.

Results

To examine whether apomorphine and MCH interacted in vitro to alter AP firing in NAshell neurons, we performed in vitro whole-cell patch-clamp electrophysiology experiments in brain slices containing NAshell medium spiny neurons. A series of 500 ms depolarizing steps of varying current intensity were applied every 30 s (see Methods) to generate AP firing. As shown in Figs. 1A and 1B, 3 µM apomorphine did not enhance NAshell firing on its own (n = 7, 0.2 ± 3.8% change in firing versus basal), although 10 µM apomorphine was sufficient to enhance firing (Fig. 1C, n = 7, 24.2 ± 8.4% change in firing versus basal, paired t-test P = 0.021 baseline versus 10 µM apomorphine). In contrast, 3 µM apomorphine significantly enhanced firing when combined with 1 µM MCH (Figs. 1A and 1B, n = 7, 24.4 ± 7.8% change in firing versus basal, P < 0.01 versus 3 µM apomorphine alone). We also found that 1 µM MCH in combination with 10 µM apomorphine significantly increased firing (Fig. 1E, n = 5, 29.5 ±6.3% change in firing versus basal, P = 0.010 paired t-test). 1 µM MCH in combination with 1 µM apomorphine did not change firing (Fig. 1D, n = 6, 0.7 ± 2.8% change in firing versus basal, P = 0.798 paired t-test), suggesting that a minimal amount of dopamine receptor activation was required to interact with MCH to increase NAshell firing. Previous studies (Chung et al., 2009; Sears et al., 2010) have shown that MCH either does not increase firing (Chung et al., 2009) or reduces firing (Sears et al., 2010) in the NAshell when applied alone; in either case it does not enhance NAshell firing when applied alone. Thus, our results suggest that MCH can interact with a sub-threshold dose of apomorphine (3 µM) to increase NAshell firing in vitro.

Since our experiments were performed in the presence of DNQX to inhibit spontaneous AMPA receptor activation and picrotoxin to inhibit spontaneous GABAA receptor activation (Hopf et al., 2003; Hopf et al., 2005), we next examined whether 3 µM apomorphine in combination with 1 µM MCH increased firing in the absence of these blockers, which might reflect more physiological conditions. As shown in Fig. 1F, apomorphine + MCH did significantly enhance firing under conditions where AMPA and GABAA receptors were not blocked (n = 5, 30.3 ± 9.6% change in firing versus basal, P = 0.030 paired t-test), suggesting that the ability of apomorphine + MCH to enhance firing persisted even when AMPA and GABAA receptors could be modulated by possible dopaminergic effects on spontaneous release (Nicola & Malenka, 1997; but see Ortinski et al., 2012) or spontaneous activity of GABA interneurons (Centonze et al., 2003).

A number of studies suggest that PKA is an central regulator of NAcb dopaminergic signaling (Svenningsson et al., 2004) including dopamine receptor enhancement of NAshell firing (Hopf et al., 2003). In agreement, the increase in firing with 3 µM apomorphine + MCH was prevented by including the PKA inhibitor peptide PKI (20 µM, Hopf et al., 2005) in the intracellular pipette solution (Fig. 1G; n = 5, 2.1 ± 5.1% change in firing versus basal, P = 0.030 vs no PKI in Fig. 1B).

We next determined whether the apomorphine + MCH increase in firing required dopamine type 1 (DA1) and/or dopamine type 2 (DA2) receptors. Application of 3 µM apomorphine + MCH did not increase firing after 10 min pre-exposure to a blocker for DA1 receptors (Fig. 2A, SCH23390 1 µM, n = 5, −3.9 ± −2.2% change in firing versus basal), a blocker for DA2 receptors (Fig. 2B, raclopride 1 µM, n = 6, −0.1 ± 8.2% change in firing versus basal) or a blocker that targets both DA1 and DA2 receptors (Fig. 2C, α-flupenthixol 2 µM, n = 4, −4.2 ± 3.4% change in firing versus basal). The increase in NAshell with 3 µM apomorphine + MCH was also prevented by pre-exposure to the MCH receptor inhibitor TPI (2 µM, Chung et al., 2009) (Fig. 2D, n = 6, 3.0 ± 2.4% change in firing versus basal), confirming the importance of MCH receptors in the action of apomorphine + MCH. A one-way ANOVA comparing the groups shown in Fig. 3 found a significant effect across groups (F(5, 29) = 4.140, P = 0.006), and post-hoc analyses showed that the 3 µM apomorphine + MCH enhancement in firing was significantly prevented by pre-exposure to any of the four receptor blockers tested here (P < 0.05). Together, these results suggest that MCH in combination with 3 µM apomorphine was able to enhance firing in NAshell neurons, and that this increase in firing required activity at both DA1 and DA2 receptors as well as MCH receptors and PKA.

Figure 3 ANOVA for dopamine receptor mediation of apomorphine/MCH enhancement of firing.

Significant effect (*P < 0.05) of MCH +apomorphine (3 µM) versus 3 µM apomorphine alone or versus MCH + 3 µM apomorphine after pre-exposure to antagonists for DA1 (SCH), DA2 (Racl), DA1/DA2 (flupen) or MCH (TPI) receptors.

To further explore the mechanism underlying the apomorphine/MCH increase in firing, we analyzed the input resistance and several aspects of the action potential waveform from all agonist conditions that enhanced action potential firing (Table 1). In agreement with our previous work (Hopf et al., 2003), increases in firing with apomorphine/MCH were not accompanied by any changes in these parameters, suggesting that the apomorphine/MCH increase in firing did not reflect alterations in inward rectifying channels or sodium or fast potassium channels active during an action potential (Hopf et al., 2003; Hopf et al., 2010b).

Table 1 Enhancement of firing with apomorphine + /− MCH was not accompanied by changes in input resistance (Rinput) or several action potential waveform parameters.

(P value for paired t-test comparisons between values at baseline and values after addition of apomorphine + /− MCH). Analyses were only performed under conditions where firing was enhanced. “Apo3 + MCH without blockers” indicates experiments where Apo3 + MCH was added in the absence of picrotoxin and DNQX. “AP” indicates action potential, and waveform parameters were determined as previously described (Hopf et al., 2003; Hopf et al., 2010b). Although there is some variability among groups in terms of basal input resistance, a one-way ANOVA found no differences in basal input resistance across the four groups tested (F(3, 20) = 1.102, P = 0.372).

Measure	Basal	Apo3 + MCH	Basal	Apo10	Basal	Apo10 + MCH	Basal	Apo3 + MCH
without blockers	
Rinput (MΩ)	174.6 ± 21.4	184.6 ± 21.0
P = 0.083	158.7 ± 28.0	164.1 ± 31.3
P = 0.345	127.8 ± 16.8	143.8 ± 18.4
P = 0.149	126.8 ± 14.8	129.9 ± 14.7
P = 0.554	
AP amplitude (mV)	80.9 ± 2.5	78.8 ± 3.6
P = 0.168	79.1 ± 1.6	78.7 ± 1.5
P = 0.583	82.4 ± 3.1	80.8 ± 4.7
P = 0.430	78.8 ± 2.5	77.4 ± 3.6
P = 0.083	
AP threshold (mV)	−45.5 ± 1.7	−46.2 ± 2.3
P = 0.343	−47.3 ± 2.3	−48.1 ± 2.5
P = 0.097	−46.2 ± 1.2	−47.7 ± 1.1
P = 0.077	−44.6 ± 2.6	−45.5 ± 2.4
P = 0.182	
AP half-width (ms)	1.85 ± 0.16	1.79 ± 0.09
P = 0.523	1.77 ± 0.09	1.83 ± 0.08
P = 0.319	1.77 ± 0.06	1.82 ± 0.09
P = 0.278	1.83 ± 0.11	1.82 ± 0.13
P = 0.983	
Peak of the fast AHP (mV)	−59.5 ± 2.3	−59.8 ± 3.0
P = 0.700	−63.1 ± 1.5	−63.3 ± 1.6
P = 0.603	−60.3 ± 1.8	−61.4 ± 1.9
P = 0.240	−58.3 ± 3.1	−58.5 ± 3.0
P = 0.449	

Discussion

Using in vitro whole-cell patch-clamp electrophysiology, we found that MCH and apomorphine in combination enhanced firing of NAshell neurons; an increase in firing was not observed with this concentration of apomorphine on its own, and MCH alone was previously shown not to increase NAshell firing (Chung et al., 2009; Sears et al., 2010). We also found that the increase in NAshell firing with apomorphine + MCH was inhibited by antagonists of either DA1, DA2, or MCH receptors or by an agent that blocks both DA1 and DA2 receptors. These results concur with our previous findings that DA1 and DA2 receptors interact to enhance NAshell activity (Hopf et al., 2003; Hopf et al., 2005; Chung et al., 2009) (see also Seif et al., 2011 in the NAcb core and Inoue et al., 2007), and suggest that apomorphine enhancement of firing required both DA1 and DA2 receptors in addition to an interaction with MCH receptors. The apomorphine + MCH increase in firing was prevented by a PKA inhibitor, consistent with previous work showing that PKA is an important mediator of NAcb dopaminergic signaling involving DA1 receptors (Svenningsson et al., 2004), including dopamine receptor enhancement of NAshell firing (Hopf et al., 2003). Our results also suggest that the NAshell may represent the brain region where lower doses of apomorphine can interact with MCH in vivo to suppress PPI (Chung et al., 2011a).

We previously showed that MCH in combination with both DA1 and DA2 receptor agonists significantly enhanced NAshell firing in vitro (Chung et al., 2009), but we did not determine whether DA1 or DA2 might itself be sufficient to interact with MCH to enhance firing. Thus, the present results with DA receptor antagonists provide new information that apomorphine activation of both DA1 and DA2 receptors was required for apomorphine to interact with MCH to enhance firing. This agrees with our previous work that dopamine enhancement of firing can be prevented by either DA1 or DA2 receptor antagonists (Hopf et al., 2003). We should note that the DA1 receptor blocker SCH23390 has been shown to directly inhibit GIRK channels (Kuzhikandathil & Oxford, 2002). However, when acting through this mechanism, SCH23390 directly depolarizes cells and enhances action potential firing; we observed no effect of SCH23390 alone on firing in a previous study (Hopf et al., 2003) or here with a shorter baseline period before adding SCH23390, suggesting that SCH23390 did not directly alter GIRK function under the conditions studied here.

Our results suggesting a DA1/DA2 interaction to increase firing also agree with a larger behavioral literature that has shown that a number of behaviors are mediated through a DA1/DA2 interaction in the NAcb (e.g. Nowend et al., 2001; Bernal et al., 2008; Shin et al., 2008; Hasbi, O’Dowd & George, 2011), and for cocaine-primed reinstatement (Schmidt & Pierce, 2006) and intra-NAshell self-administration of dopamine receptor activators (Ikemoto et al., 1997), DA1 and DA2 receptors interact cooperatively to support behavior. In contrast, some behaviors are regulated differentially by DA1 and DA2 receptors in the NAcb (Goto & Grace, 2005; Pezze, Dalley & Robbins, 2007; Haluk & Floresco, 2009); at a cellular level, DA1 and DA2 receptors can have divergent effects, e.g., on glutamate receptor function (Georgescu et al., 2005; Goto & Grace, 2005; Sears et al., 2010; Wang et al., 2012). Here, we found that apomorphine + MCH increased firing when AMPA and GABAA receptors were not blocked pharmacologically, suggesting that the increase in firing still occurred in the presence of any possible dopaminergic effects on glutamate or GABAA receptors due to spontaneous release (Nicola & Malenka, 1997; but see Ortinski et al., 2012) or spontaneous activity of GABAergic interneurons (Centonze et al., 2003). Although the mechanism through which DA1 and DA2 receptors in the NAcb might interact positively under some conditions remain unclear, studies have described a mechanism requiring DA1-DA2 heterodimers (Hasbi, O’Dowd & George, 2011) or G protein beta-gamma subunits (Hopf et al., 2003; Inoue et al., 2007) and PKA (here; Hopf et al., 2003; Inoue et al., 2007).

We should also note that studies of MCH alone on NAshell activity have yielded somewhat mixed results. We found that MCH alone did alter neuronal firing or the levels of phospho-DARPP but that MCH in combination with dopamine receptor agonists enhanced both measures (Chung et al., 2009). In contrast, Sears and colleagues (Sears et al., 2010) found that MCH alone can inhibit firing and phospho-GluR1 in the NAshell. Although the reason for these mixed results is not completely apparent, it is possible that MCH would be able to modulate firing when combined with activation of dopamine receptors but have different effects on firing when applied alone. Such divergent effects of receptor agonists alone versus in combination with other types of receptor agonists has previously been shown, for example as shown with adenosine/cannabinoid receptor interactions (Tebano, Martire & Popoli, 2012).

Conclusion

We found that MCH interacted with subthreshold levels of the dopamine receptor agonist apomorphine to enhance firing of NAshell neurons in vitro. Apomorphine + MCH enhancement of NAshell firing required both DA1 and DA2 receptors as well as MCH receptors. Our in vitro results parallel what is observed in vivo where MCH enhances the ability of lower doses of apomorphine to suppress PPI, an animal model for some aspects of schizophrenia in humans. Thus, the NAshell may represent the brain region where MCH and apomorphine interact to regulate the expression of PPI and thus could regulate some symptoms of human schizophrenia. Apomorphine within the NAshell can also reduce withdrawal-related anxiety (Radke & Gewirtz, 2012), suggesting that this mechanism might be relevant to other psychiatric diseases such as depression which are modulated through MCH (Georgescu et al., 2005; Nestler & Carlezon, 2006; Shirayama & Chaki, 2006; Chung et al., 2009; Morganstern et al., 2010; Sears et al., 2010; Mul et al., 2011; Chung et al., 2011a; Chung et al., 2011b).

Additional Information and Declarations

Competing Interests

Author Contributions

Animal Ethics

Olivier Civelli is an Academic Editor for PeerJ. The authors declare no other competing interests.

F Woodward Hopf and Taban Seif conceived and designed the experiments, performed the experiments, analyzed the data, wrote the paper.

Shinjae Chung conceived and designed the experiments.

Olivier Civelli conceived and designed the experiments, wrote the paper.

The following information was supplied relating to ethical approvals (i.e. approving body and any reference numbers):

“The animal use in this study was performed in accordance with Institutional Animal Care and Use and National Institutes of Health Guidelines and under the guidance and approval of the IACUC Committee of the Ernest Gallo Clinic and Research Center.” The IACUC approval number is 11.09.244.

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
