# Peer review of "MCH and apomorphine in combination enhance action potential firing of nucleus accumbens shell neurons in vitro"

_PeerJ, doi:10.7717/peerj.61_

## Round 0.1 · original submission · Major Revisions

Your study is an interesting example of interactions between a small molecule- and a neuropeptide-system in an important field - addiction. The reviewers express relevant critique, and we hope that you will be able to
respond in a convincing way.

Reviewer 1 ·

Basic reporting

No Comments

Experimental design

No Comments

Validity of the findings

The in vitro data here is interesting with consistency of in vivo data (PLOS One 2011). However, this pharmacological experiment is a kind of “repeated” one, since the author essentially performed the identical experiments with that of Fig. 2A in PNAS (2009). Apomorphine is DA1/DA2 dopamine receptor activator, so simultaneous addition of D1 agonist and D2 agonist logically cause the same effect as the addition of apomorphine.

Additional comments

Although the basic finding is potentially interesting, the critical data is essentially overlapped with their previous findings and the present manuscript is with less data. The authors need further solid pharmacological data, for example, firing of Apo1+MCH, Apo10+MCH or the degree of phospholylation of DARPP-32 in slices, before this paper might be ready for publication.

·

Basic reporting

In the article entitle “MCH and apomorphine in combination enhance action potential firing of nucleus accumbens shell neurons in vitro », Hopf and colleagues present an ex-vivo electrophysilogical characterization of the potentiating effects of MCH on the activating properties of DA agonist on medium spiny neurons of the NAshell. The study is well conducted and data are clear and of a real interest. The paper is well written and easy to read. Nevertheless, from my point of view, some points and/or explanations should be added.

In the introduction, the PPI protocol should be better define to facilitate understanding from non-specialists.

Finally, the implications in schizophrenia of the NAshell, mostly known as a key structure of the brain reward pathway, should perhaps be a little more discussed.

Experimental design

The study was conducted in the presence of AMPA and GABA-A receptors blockers, which is not the case to my knowledge for in vivo studies. Because of the strong interaction between the DA systems and, in particular, the GABA interneurons in the NAshell, the effects of the co-application of MCH and apomorphine in standard conditions (without blockers) may be really interesting. Authors should also take into account that MCH neurons are mainly GABAergic.

Validity of the findings

As written by the authors, the interaction between DA1 and DA2 pathways in the NAshell may be relatively specific to this structure, but also controversial. Nevertheless, because of the divergence in the intracellular coupling of DA1 and DA2, it still strange to me to have such a cooperative effect. More specifically, it as been shown that the DA1 “antagonist” SCH23390 may block some component of the cellular response classically inhibited by DA1 such as some GIRK channels. As a consequence, I think that the authors should try another DA1 antagonist, perhaps a bit less specific of DA1, but less “dirty” or, because of the cooperation suggested by the results, a mixed antagonist of DA1 and DA2 and compared the results to the “specific” DA2 antagonist raclopride. A bit of intracellular “mechanistic” may also clarify these points.

It may be really interesting to briefly present the electrical properties of the patched cells, instead making reference to a previous study (even if from the same author).

Additional comments

As a conclusion, this study is well conducted and highlights a potential new role of MCH/DA system interaction in the NAshell, in the potential treatment of psychiatric disorders. In the case of convincing answers or argumentation in regard of the two major critics I formulate, this study may be published in PeerJ.

---

## Round 0.2 · accepted · Accept

The authors have made a thorough revision according to the reviewers' suggestions, including expanding the introduction and carrying out a number of new experiments. The manuscript now represents a clear progress compared to their previous study. Thus, the evidence for an important MCH - DA interaction has been distinctly strengthened.